# Convolution Neural Networks for the Automatic Segmentation of 18F-FDG PET Brain as an Aid to Alzheimer's Disease Diagnosis

**Elena Pasini** [1], **Dario Genovesi** [2], **Carlo Rossi** [3], **Lisa Anita De Santi** [4], **Vincenzo Positano** [2], **Assuero Giorgetti** [2] and **Maria Filomena Santarelli** [1,*]

1  Institute of Clinical Physiology, National Research Council (CNR), 56124 Pisa, Italy; epasini@ifc.cnr.it
2  Fondazione G. Monasterio CNR-Regione Toscana, 56124 Pisa, Italy; genovesi@ftgm.it (D.G.); positano@ftgm.it (V.P.); assuero.giorgetti@ftgm.it (A.G.)
3  Neurology Unit, Hospital of Pontedera, 56025 Pontedera, Italy; carlo78rossi@gmail.com
4  Dipartimento di Ingegneria dell'Informazione (DII), Pisa University, 56126 Pisa, Italy; lisa.desanti@phd.unipi.it
*  Correspondence: santarel@ifc.cnr.it

**Abstract:** Our work aims to exploit deep learning (DL) models to automatically segment diagnostic regions involved in Alzheimer's disease (AD) in 18F-fluorodeoxyglucose positron emission tomography (18F-FDG PET) volumetric scans in order to provide a more objective diagnosis of this disease and to reduce the variability induced by manual segmentation. The dataset used in this study consists of 102 volumes (40 controls, 39 with established Alzheimer's disease (AD), and 23 with established mild cognitive impairment (MCI)). The ground truth was generated by an expert user who identified six regions in original scans, including temporal lobes, parietal lobes, and frontal lobes. The implemented architectures are the U-Net3D and V-Net networks, which were appropriately adapted to our data to optimize performance. All trained segmentation networks were tested on 22 subjects using the Dice similarity coefficient (*DSC*) and other similarity indices, namely the overlapping area coefficient (*AOC*) and the extra area coefficient (*EAC*), to evaluate automatic segmentation. The results of each labeled brain region demonstrate an improvement of 50%, with *DSC* from about 0.50 for V-Net-based networks to about 0.77 for U-Net3D-based networks. The best performance was achieved by using U-Net3D, with *DSC* on average equal to 0.76 for frontal lobes, 0.75 for parietal lobes, and 0.76 for temporal lobes. U-Net3D is very promising and is able to segment each region and each class of subjects without being influenced by the presence of hypometabolic regions.

**Keywords:** 18F-FDG PET; deep learning; segmentation; convolutional neural network; Alzheimer's disease

## 1. Introduction

Alzheimer's disease (AD) is a neurodegenerative disorder that leads to memory loss and other functional impairments and is the most common form of dementia [1]. Currently, AD is diagnosed via clinical evaluation and neuropsychological tests; however, clinical tests show several limitations for early detection and the objectivity of final diagnosis.

Diagnostic neuroimaging tests, typically positron emission tomography (PET) or magnetic resonance imaging (MRI), play essential roles in the investigation of dementia. MRI is used to analyze the structure of the brain, detecting the presence of markers such as gray matter atrophy and ventricular enlargement, which are established markers for pathology, and 18F-fluorodeoxyglucose (18F-FDG) PET reveals the pattern of neuronal uptake and cerebral distribution, which is also a discriminating factor for AD.

In recent studies, 18F-FDG PET has been proven to be a promising neuroimaging tool in the diagnosis of AD because it reveals the cerebral metabolic rate of glucose, which

is an indicator of neuronal activity. Several studies have shown that cerebral metabolic alterations correlate with the clinical manifestation of AD symptoms [2]. The classic pattern of impaired metabolism consists of the involvement of temporal lobes and parietal lobes, and then, in more advanced Alzheimer's disease, hypometabolism extends to involve the frontal lobe [3].

An objective analysis of 18F-FDG PET images requires the 3D segmentation of specific brain regions involved in the AD progression to quantify the metabolic activity from the PET signal value. Manual or semi-automatic 3D segmentation techniques are widely used in clinical practice but are associated with high processing time and inter-and intra-observer variability. PET images are influenced by intrinsic and extrinsic factors that make the generalization of PET segmentation methods more challenging [4]: (i) low resolution and high smoothing, which reduce the contrast between objects in the image; (ii) large variability in the shape of texture, due to the presence of the pathology, can be present; (iii) image noise can reduce the feasibility of segmentation methods based on the standardized uptake value (SUV) or on the intensity of the initial "seed" location defined in some algorithms, such as in region-growing methods.

In the literature, there are effective PET segmentation algorithms for specific clinical uses, but they require expert users' interaction, with a high impact on segmentation times and repeatability. Therefore, the development of fully automated methods is highly desirable.

Recently, machine learning technologies have been applied in medical image processing, based on fully convolutional networks [5] or other supervised networks for 3D volumes [6].

In particular, convolutional neural networks (CNNs) have gained widespread attention for the implementation of image segmentation: With the definition of new, specialized architectures for 3D volume segmentation (e.g., U-Net3D [7] and V-Net [8]), it is currently possible to realize semantic segmentation in volumetric CT and MR images for the diagnosis of brain tumors [9], liver tumor [10], oropharyngeal cancer [11], lung nodules [12], and bone structure [13]. However, it can be useful to use this kind of architecture on different imaging techniques, such as PET 3D. In the last two years, deep learning (DL) techniques have been developed for tumor segmentation on PET-CT images [14]. Recently, CNN architectures have been proposed to obtain the automatic classification of AD and mild cognitive impairment (MCI) in diagnosing AD from 18F-FDG PET imaging data [15]. However, as far as we know, no models are proposed for automatically identifying diagnostic regions on the brain's 18F-FDG PET 3D data volumes to help AD diagnosis. This work aims to provide automatic techniques to support a more objective and potentially accurate diagnosis that could reduce the problems of variability and subjectivity typical of manual segmentation. In fact, from the volumetric segmentation obtained by the proposed networks, with the values of the signal within the regions proportional to metabolic activity, quantitative indices that determine the presence of the pathology should be automatically derived. These quantitative parameters can help the physician to classify subjects and, eventually, to follow the evolution of the disease over time.

In this work, different solutions are proposed, based on U-Net3D and V-Net convolutional neural network (CNN) architectures, which are the most well-known DL architectures in medical image segmentation. The architectures were transformed and personalized to optimize the performance in time and accuracy of automatic segmentation for 18F-FDG PET brain images.

## 2. Materials and Methods

### 2.1. Study Population

Data used in the preparation of this article were obtained from our institutional database and concerned 102 subjects with clinical suspicion of dementing disorders. A follow-up of at least 24 months was used to clinically categorize the subjects in 40 controls, 39 patients who met the criteria for probable Alzheimer's disease of the National Institute

of Neurological and Communicative Disorders and Stroke and the Alzheimer's Disease and Related Disorders Association [16], and 23 patients who met criteria for MCI [17].

### 2.2. 18F-FDG PET 3D Data Acquisition and Preprocessing

PET/CT scans were obtained using a Discovery VCT 64-slice tomography scanner (GE Healthcare, Milwaukee, WI, USA). Three-dimensional (3D) list-mode PET acquisition was performed following EANM procedure guidelines for PET brain imaging [18]. Briefly, patients were studied in fasting conditions, and blood glucose levels were checked prior to radiotracer administration to exclude hyperglycemia. Blood glucose levels were considered acceptable if <160 mg/dL. Then, 18F-FDG was intravenously injected using a fully automated PET infusion system (MEDRAD® Intego; 18F-FDG range 185–260 Mbq). A three-dimensional (3D) list-mode PET/CT acquisition was performed for 10 min, about 30–40 min after the radiotracer injection. Then, 3D images were reconstructed from the list-mode data using the ordered-subset expectation maximization (OSEM) iterative technique, with 3 iterations and 21 subsets; thus, a $128 \times 128 \times 47$ voxels matrix was obtained. The obtained 3D images are axial slices, showing cross-sections of the brain with 3.3 mm of thickness, for a field of view of 150 mm.

Then, the acquired 3D 18F-FDG datasets were subjected to a preprocessing phase that consisted of choosing consecutive slices, which include those regions where the presence of Alzheimer's disease typically occurs; obtaining sub-volumes of size $128 \times 128 \times 20$; and the manual definition of ground-truth masks by an expert using a MIPAV segmentation tool [19]. Moreover, to evaluate the reproducibility of this method and the inter-observer variability, a second expert manually segmented the same data volumes. Both observers were blinded to each other's results.

The regions of interest (ROIs) on each image were labeled as follows: background (label 0), the right frontal lobe (label 1), the left frontal lobe (label 2), the right parietal lobe (label 3), the left parietal lobe (label 4), the right temporal lobe (label 5), and the left temporal lobe (label 6). In Figure 1, the examples of PET slices (left) and manually defined ROIs (right) relevant to the six anatomical regions are shown.

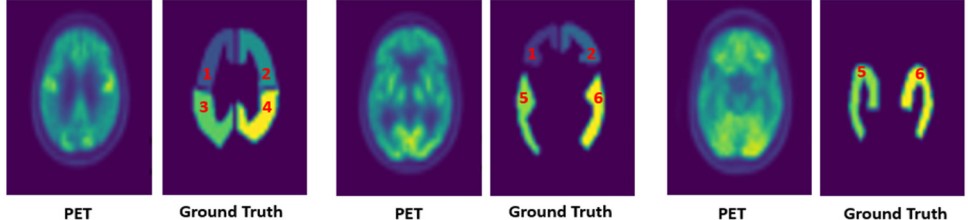

**Figure 1.** PET image (**left**) and manual segmentation (**right**) from three representative slices: right frontal lobe (label 1), left frontal lobe (label 2), right parietal lobe (label 3), left parietal lobe (label 4), right temporal lobe (label 5), and left temporal lobe (label 6).

Each brain region occupied several axial slices, and the number of such slices varied between the different subjects: in a data volume, on average, about 10% of the slices included only the temporal lobe, 40% of the slices covered the temporal and frontal lobes, and 50% covered the frontal and parietal lobes.

Successively, in order to use the images as input for the proposed CNN, the entire dataset was randomly split into groups as follows: 80 volumes for use during network training, which were, in turn, divided into 60 3D data for the train and 20 for validation, and 22 volumes for testing the trained net.

### 2.3. Proposed Architectures

In order to evaluate, among the different families of CNN, which ones could be optimal for our purposes, in the present work, we considered five different models: three of them

exploit the properties of the U-Net3D model, while the other two are based on the V-Net model. Their characteristics are described in more detail below.

### 2.3.1. U-Net3D

This paper provides image segmentation methods based on U-Net3D architectures (Figure 2a) [7]. These networks' structure is divided into two symmetrical paths: an encoder path that captures features associated with the context of images and a decoder path that constructs segmentation maps from the encoded features.

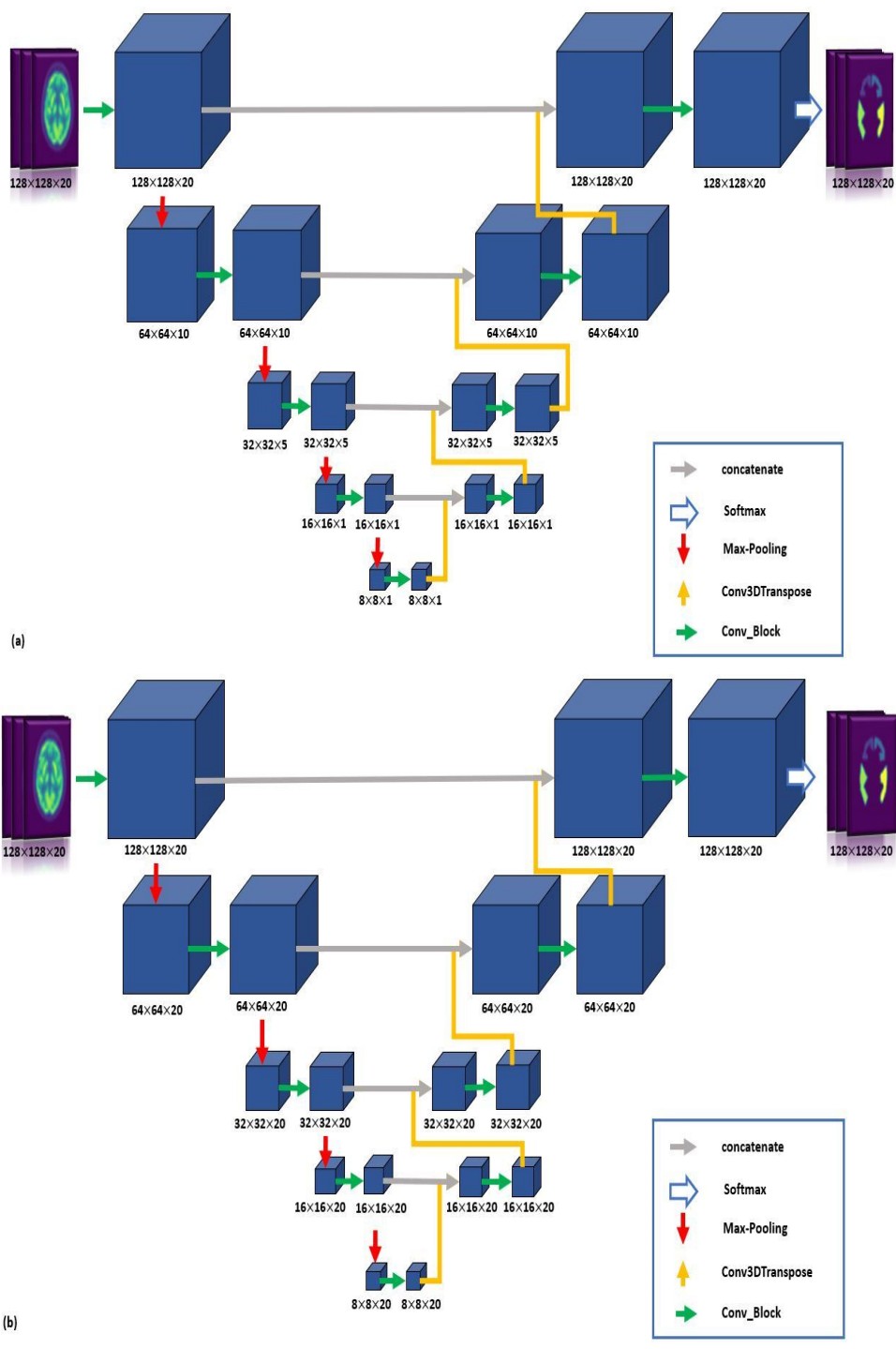

**Figure 2.** *Cont.*

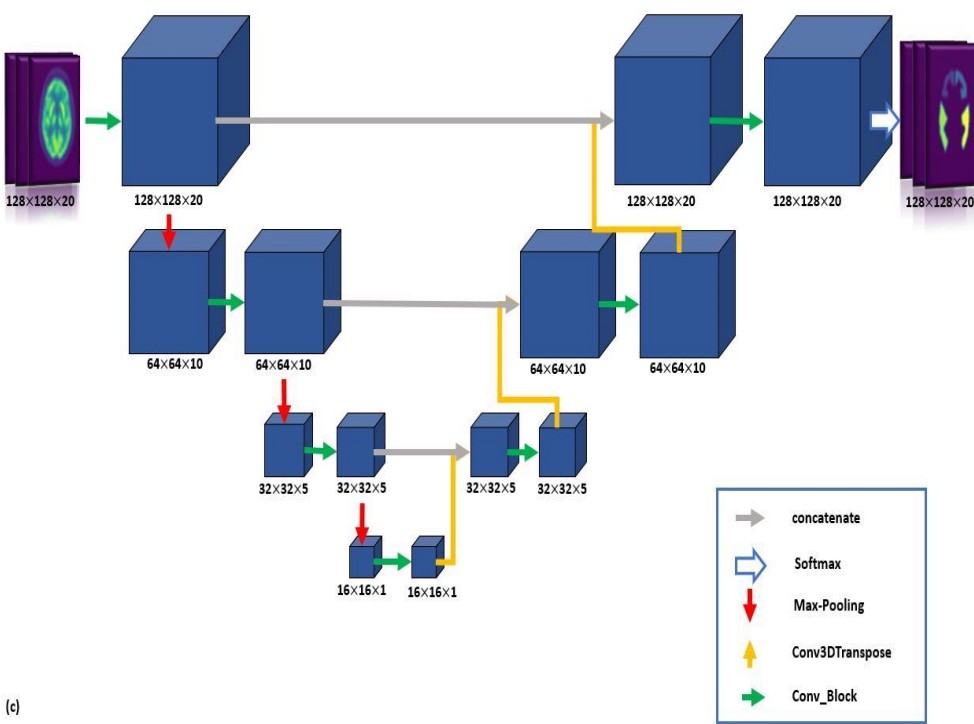

**Figure 2.** U-Net3D architectures: (**a**) U-Net3D; (**b**) U-Net3D-NoMaxPoolingThirdDimension; (**c**) U-Net3D-TwoLevel. The arrows correspond to operations or functions applied to the images; the blue cubes are the image volumes; numbers under the cubes are volume sizes.

The encoder path includes several layers, whose number changes from model to model, consisting of the following features:

- **Conv_blocks,** which are characterized by (i) two consecutive $3 \times 3 \times 3$ filter blocks; (ii) batch normalization [20], which normalizes the inputs of hidden layers (subdivided in mini-batch) to make the net training faster and more stable; and (iii) *ReLu* activation function that directly gives the input as output if the input is positive, and zero otherwise:

$$ReLu(q) = \begin{cases} 0, & if\ q \leq 0 \\ q, & otherwise \end{cases} \tag{1}$$

- **Max-pooling** block to halve the image size.

The number of filters is constant in each layer and doubles between adjacent layers, to only focus on the features of interest.

The decoder path, based on the same number of layers of the encoder path, consists of Conv_blocks, which take as input the Convolution 3D Transpose of the previous layer output concatenated with the output of the corresponding layer of the encoder path.

The last operation of the net consists of a Softmax activation function (blue-bordered arrow, in the top right of Figure 2a–c) which assigns a class (label) to each pixel and produces segmented volumes with the same resolution of the input volume.

The network manages $N \times N \times S \times 1$ volumes as input and produces $N \times N \times S \times C$ volumes as output, where $N \times N$ (128 $\times$ 128) is the image size, S = 20 is the number of slices, and C = 7 is the number of classes, i.e., labels.

We analyzed the results obtained using three different models of the U-Net3D architecture trained for cerebral segmentation:

- **U-Net3D**, which is the original architecture composed of three encoder and decoder layers (Figure 2a);
- **U-Net3D-NoMaxPoolingThirdDimension,** where, for each layer, MaxPooling is not applied to the third dimension (Figure 2b). The network acquires more information

for each layer, and a higher number of parameters must be estimated during the training phase;

- **U-Net3D-TwoLevel** (Figure 2c), which is a model characterized by eliminating one layer from the original architecture, reducing the number of parameters.

### 2.3.2. V-Net

The v-Net architecture was adopted for the first time by [8] and can handle the entire volume of data using volumetric convolutions.

This architecture (see Figure 3a) is relatively similar to the U-Net3D and, as U-Net, consists of a compression path (encoder path), followed by an expansion path (decoder path). Encoder and decoder paths have the same number of layers, and the number of layers in the network depends on the specific architecture adopted.

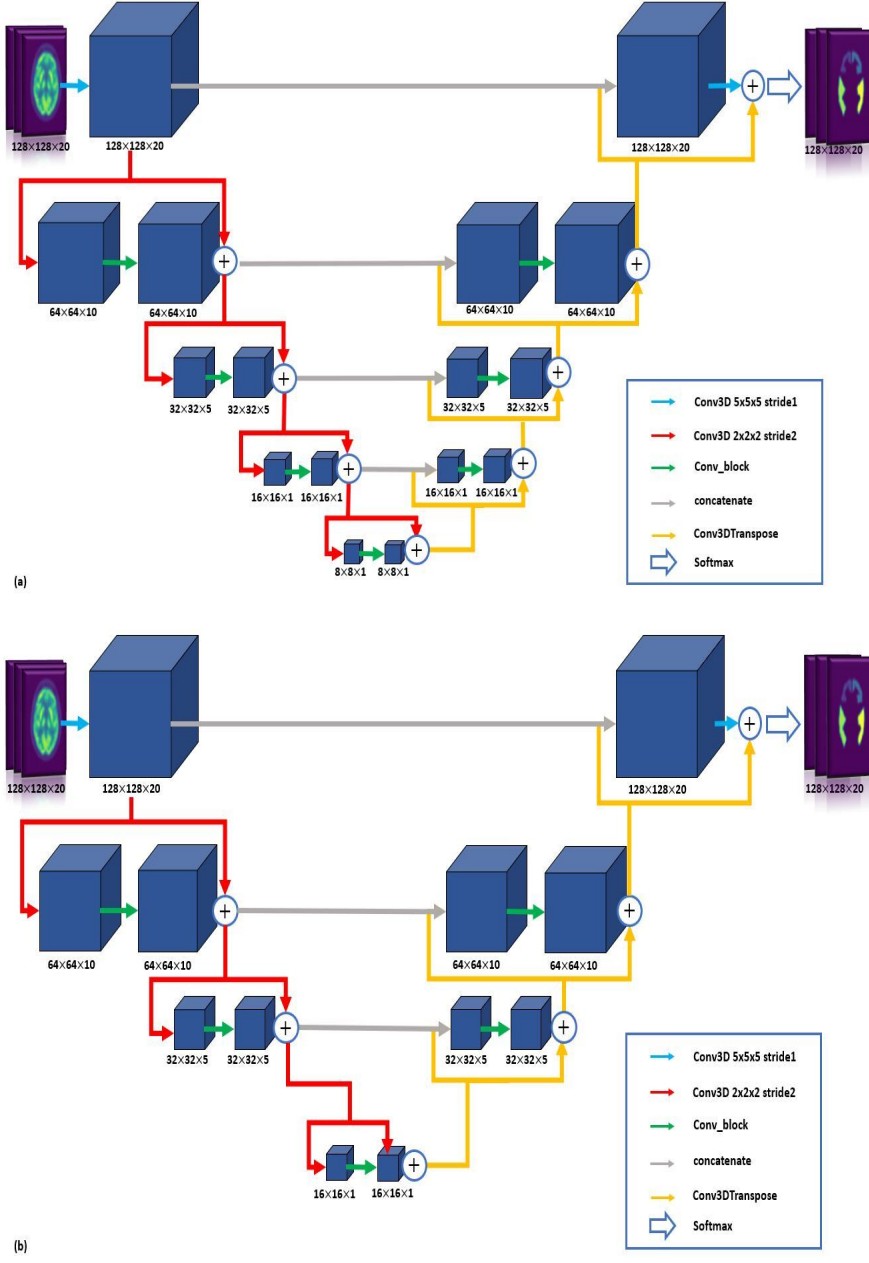

**Figure 3.** V-Net architectures: (**a**) V-Net, (**b**) V-Net-TwoLevel. The arrows correspond to operations or functions applied to the images; the blue cubes are the image volumes; numbers under the cubes are volume sizes.

The encoder path consists of the following features:

- **Conv_block** composed of two successive $5 \times 5 \times 5$ filters, followed by the same operations as those described for the U-Net architectures, i.e., batch normalization and ReLu activation (see Section 2.3.1);
- **Conv3D** of size $2 \times 2 \times 2$ and stride 2 to halve the size of the image.

In the encoder path, the number of filters is doubled going down from one layer to the next, to extract the features of interest.

Similar to U-Net, the decoder path for V-Net was also based on the same number of layers as those of the encoder, consisting of a convolution block having as input the Convolution 3D Transpose from the previous layer concatenated with the output of the same layer of the encoder path. An additional operation in the V-Net architecture, which is not present in the U-Net model, is performed on each convolutional block: In the compression (and the expansion) path, the block learns a residual function owing to the sum of the higher (lower) level output with the Conv_block output (blue-circled 'plus' symbol in Figure 3a,b).

The last operation in the V-Net model, similar to the U-Net model, is a Softmax activation function (blue-bordered arrow, in the top right of Figure 3a,b).

The network takes blocks of sizes $N \times N \times S \times 1$ as input, with $N = 128$ and $S = 20$, and returns $N \times N \times S \times C$ as output, with $C = 7$.

Two different models for V-Net architecture were trained:

- **V-Net**, with an architecture similar to that of the original (Figure 3a) and composed of three layers;
- **V-Net-TwoLevel**, which is a model characterized by the elimination of one layer from the original architecture (Figure 3b).

### 2.4. Data Augmentation

Data augmentation aims to improve the network performance by intentionally producing more training data from the original one and reducing overfitting probability. We applied simple operations such as rotation (range $[-10, 10]$; random degrees) and shift ($-10,10$ pixels). After that, we obtained a total of 360 volumes for training.

### 2.5. Training

All the proposed neural network architectures were implemented using Python utilities with Keras framework [21] on TensorFlow [22] backend. The training was performed on a personal computer that includes a GPU Nvidia, model TITAN XP, with 12 Gb of memory.

The training performance was evaluated as a loss function, with the "multiclass Dice loss" expressed as

$$DiceLoss = 1 - \frac{2}{|K|} \sum_{k \in K} \frac{\sum_i u_{i,k} v_{i,k}}{\sum_i u_{i,k}^2 + \sum_i v_{i,k}^2} \tag{2}$$

where $u$ is the Softmax output of the network, and $v$ is the ground-truth segmentation map; $i$ values are voxels in the training patch, and $k$ values are the classes (seven in our case). The values $u_{i,k}$ and $v_{i,k}$ denote the Softmax output and the ground truth for class $k$ at voxel $i$, respectively.

The metric was set to be the Dice coefficient (or Dice score), which is 1-*DiceLoss*.

The training activity, for each architecture, was executed using an Adam optimizer with decay rate and learning rate values that were optimized using the Optuna software framework [23], together with other hyperparameters shown in Table 1.

To generalize the network performance and evaluate its reliability, cross-validation [24] was executed 5 times with a random assignment of dataset volumes to training (360 volumes, after data augmentation) and validation (20 volumes).

**Table 1.** Hyperparameters setting for each network using Optuna.

|  | Batch-Size | Learning-Rate | Decay-Rate | Dropout |
|---|---|---|---|---|
| U-Net3D | 14 | 0.07 | $9.95 \times 10^{-5}$ | 0.26 |
| U-Net3D-NoMaxPoolingThirdDim | 17 | 0.04 | $8.46 \times 10^{-5}$ | 0.33 |
| U-Net3D-TwoLevel | 2 | 0.07 | $7.02 \times 10^{-5}$ | 0.41 |
| V-Net | 2 | $2.29 \times 10^{-4}$ | $4.43 \times 10^{-6}$ | 0.17 |
| V-Net-TwoLevel | 4 | $2.29 \times 10^{-4}$ | $4.43 \times 10^{-6}$ | 0.17 |

### 2.6. Evaluation Methods

### 2.6.1. Dice Similarity Coefficient

*Dice similarity coefficient (DSC)* can be used to measure the coincidence degree between the segmentation result and the ground truth. Its value ranges from 0 to 1. The closer to 1, the better the pixel classification effect of the model. The definition is as follows:

$$DSC = \frac{2 \times |X \cap Y|}{|X| + |Y|} \quad (3)$$

where $X$ and $Y$, respectively, represent the ground-truth segmentation results and the segmentation results obtained by the proposed method. $|.|$ denotes the cardinality of the set of pixels.

### 2.6.2. Other Coefficients

Further indices were defined to assess which architectures best perform for the segmentation of brain regions. These indices allow evaluating the goodness of the segmentation for each brain region relative to the ground truth.

- *Overlapping area coefficient (AOC)*

$$AOC = \frac{|X \cap Y|}{|X|} \quad (4)$$

This coefficient takes on the value 1 if the output segmentation completely overlaps with the expected one.

- *Extra area coefficient (EAC)* is an index that allows evaluating the area added by the automatic segmentation and is defined as

$$EAC = \frac{|Y - (X \cap Y)|}{|X|} \quad (5)$$

This coefficient assumes a value of 0 if the output image coincides with the overlapping regions.

### 3. Results

### 3.1. Cross-Validation

The quantitative cross-validation results are summarized as mean and standard deviation (SD) values in Table 2. The best result in the analysis is shown in bold.

**Table 2.** Cross-validation results: *DSC* values obtained from each model in training and validation datasets.

|  | *DSC* | |
|---|---|---|
|  | **Training** | **Validation** |
| U-Net3D | **0.83 ($\pm$0.03)** | **0.83 ($\pm$0.02)** |
| U-Net3D-NoMaxPoolingThirdDim | 0.79 ($\pm$0.04) | 0.76 ($\pm$0.04) |
| U-Net3D-TwoLevel | 0.80 ($\pm$0.05) | 0.80 ($\pm$0.08) |
| V-Net | 0.46 ($\pm$0.03) | 0.51 ($\pm$0.03) |
| V-Net-TwoLevel | 0.50 ($\pm$0.02) | 0.57 ($\pm$0.03) |

### 3.2. Test Dataset Analysis

The described metrics were used to evaluate the segmentation of the test dataset (22 subjects), including *DSC* and other indices (*AOC* and *EAC*).

Figure 4 shows the comparison of the ground truth (second column) relative to the proposed approaches. Original PET images (first column) are attributed to a control subject, and three slices are shown (three rows in Figure 4), to display all the anatomical regions involved in the segmentation.

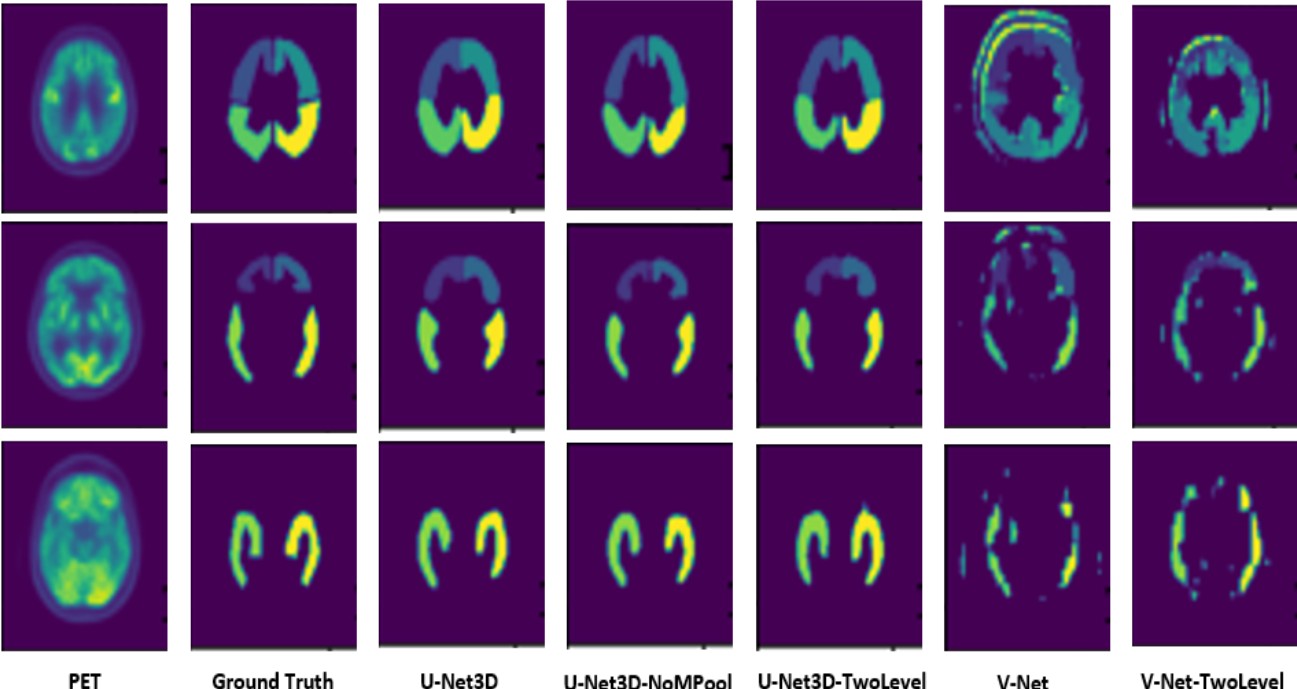

**Figure 4.** Segmentation results from a normal subject's volume. First column: 18F-FDG slices; second column: ground truth; third–fifth columns: results of the automatic segmentation using the proposed models.

In Table 3, the mean values (±SD) of the *DSC*, *AOC*, and *EAC* indices are shown; the index values shown are attributed to each brain region and each proposed model. In the table, the best index values for each model and each region are highlighted in bold.

From the results shown in Figure 4 and Table 3, the U-Net3D and U-Net3D-TwoLevel architectures were found to be the best among those proposed. Although there was not a statistically significant difference between the two models, the mean values of the indices were slightly better for U-Net3D; for this reason, U-Net3D was chosen as the best model, and its results are shown below.

**Table 3.** The results of *DSC*, *AOC*, and *EAC* indices on test dataset.

| | DSC | | | | | | AOC | | | | | | EAC | | | | | |
|---|---|---|---|---|---|---|---|---|---|---|---|---|---|---|---|---|---|---|
| | Lobe Frontal | | Lobe Parietal | | Lobe Temporal | | Lobe Frontal | | Lobe Parietal | | Lobe Temporal | | Lobe Frontal | | Lobe Parietal | | Lobe Temporal | |
| | Right | Left | Right | Left | Right | Left | Right | Left | Right | Left | Right | Left | Right | Left | Right | Left | Right | Left |
| U-Net3D | **0.74** (±0.10) | **0.77** (±0.09) | **0.76** (±0.10) | **0.74** (±0.08) | **0.76** (±0.10) | **0.77** (±0.10) | **0.75** (±0.12) | **0.75** (±0.11) | **0.73** (±0.09) | **0.73** (±0.09) | **0.71** (±0.10) | **0.71** (±0.09) | 0.42 (±0.15) | 0.39 (±0.12) | 0.35 (±0.17) | 0.33 (±0.16) | 0.31 (±0.15) | 0.31 (±0.14) |
| U-Net3D-NoMaxPoolingThirdDim | 0.70 (±0.14) | 0.73 (±0.11) | 0.74 (±0.11) | 0.73 (±0.11) | 0.74 (±0.11) | 0.73 (±0.11) | 0.64 (±0.15) | 0.65 (±0.13) | 0.66 (±0.12) | 0.66 (±0.11) | 0.66 (±0.11) | 0.66 (±0.11) | 0.32 (±0.23) | 0.30 (±0.22) | 0.30 (±0.21) | 0.29 (±0.19) | 0.29 (±0.18) | 0.28 (±0.17) |
| U-Net3D-TwoLevel | **0.75** (±0.08) | **0.75** (±0.09) | **0.77** (±0.08) | **0.77** (±0.07) | **0.75** (±0.09) | **0.77** (±0.09) | 0.68 (±0.09) | 0.68 (±0.09) | 0.71 (±0.08) | 0.72 (±0.07) | 0.70 (±0.09) | 0.70 (±0.09) | **0.28** (±0.19) | **0.29** (±0.21) | **0.29** (±0.17) | **0.29** (±0.15) | **0.28** (±0.13) | **0.26** (±0.11) |
| V-Net | 0.55 (±0.11) | 0.54 (±0.13) | 0.55 (±0.09) | 0.54 (±0.09) | 0.57 (±0.11) | 0.57 (±0.16) | 0.59 (±0.13) | 0.56 (±0.12) | 0.59 (±0.11) | 0.57 (±0.10) | 0.56 (±0.10) | 0.57 (±0.11) | 0.65 (±0.47) | 0.57 (±0.41) | 0.60 (±0.31) | 0.56 (±0.25) | 0.51 (±0.21) | 0.49 (±0.17) |
| V-Net-TwoLevel | 0.50 (±0.11) | 0.52 (±0.10) | 0.50 (±0.07) | 0.49 (±0.07) | 0.53 (±0.11) | 0.51 (±0.13) | 0.48 (±0.12) | 0.49 (±0.09) | 0.50 (±0.07) | 0.50 (±0.06) | 0.48 (±0.07) | 0.48 (±0.07) | 0.46 (±0.33) | 0.48 (±0.31) | 0.48 (±0.23) | 0.49 (±0.19) | 0.44 (±0.15) | 0.41 (±0.13) |

For a global view of the performance of the U-Net3D model, in Figure 5, *DSC*, *AOC*, and *EAC* values from each anatomic label are shown as boxplots.

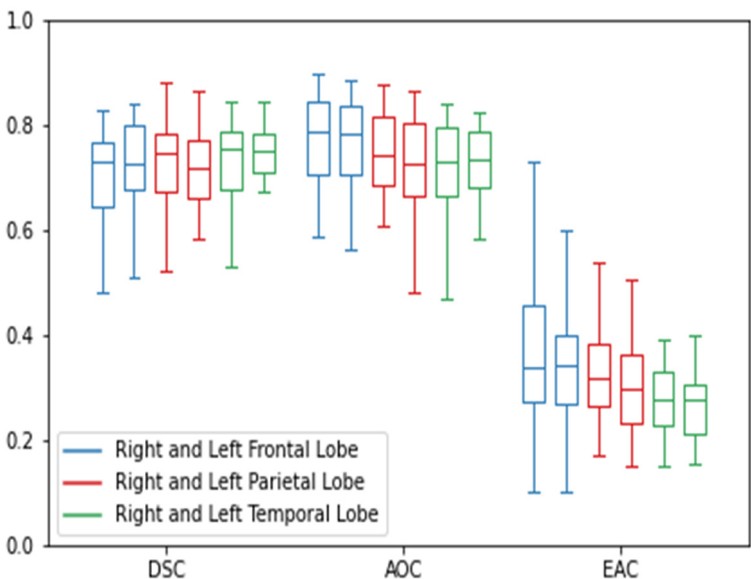

**Figure 5.** U-Net3D results of *DSC*, *AOC*, and *EAC* indices on test dataset.

Figure 6 shows an example of the results obtained from a control subject (Figure 6a), an MCI (Figure 6b), and an AD (Figure 6c): It highlights a comparison of the ground truth and prediction images for three typical types of subjects. In each row, slices are shown that include the anatomical regions of interest (i.e., frontal lobes, parietal lobes, and temporal lobes). The first column shows the three 18F-FDG- PET slices; the second column shows the relevant ground truth obtained from manual segmentation by an expert, and the third column shows the automatic segmentation obtained using the U-Net3D neural network from the test dataset.

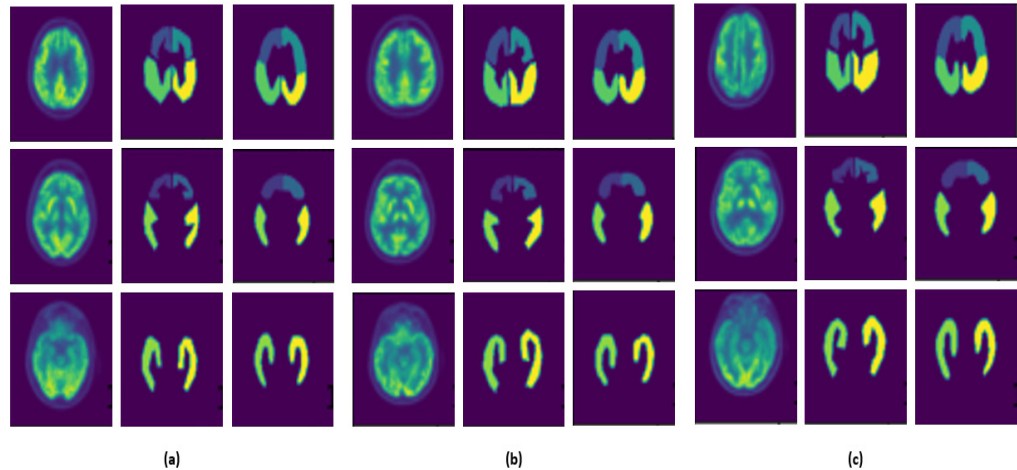

**Figure 6.** Comparison between 18F-FDG PET slices for three example subjects (first column), ground truth (second column), and automatic segmentation via U-Net3D (third column) for (**a**) control patient, (**b**) MCI patient, and (**c**) AD patient.

*3.3. Inter-Observer Variability*

Table 4 shows the mean (±SD) of the *DSC*, *AOC*, and *EAC* indices obtained from the comparison between the manual segmentation performed by the first and second experts.

**Table 4.** Results of inter-observer variability.

| | DSC | | | | | | AOC | | | | | | EAC | | | | | |
|---|---|---|---|---|---|---|---|---|---|---|---|---|---|---|---|---|---|---|
| | Lobe Frontal | | Lobe Parietal | | Lobe Temporal | | Lobe Frontal | | Lobe Parietal | | Lobe Temporal | | Lobe Frontal | | Lobe Parietal | | Lobe Temporal | |
| | Right | Left | Right | Left | Right | Left | Right | Left | Right | Left | Right | Left | Right | Left | Right | Left | Right | Left |
| Variab. Inter-Observ. | 0.61 (±0.10) | 0.62 (±0.09) | 0.62 (±0.07) | 0.61 (±0.10) | 0.64 (±0.11) | 0.62 (±0.11) | 0.61 (±0.12) | 0.62 (±0.10) | 0.64 (±0.10) | 0.66 (±0.10) | 0.64 (±0.08) | 0.64 (±0.07) | 0.33 (±0.21) | 0.32 (±0.21) | 0.28 (±0.15) | 0.27 (±0.13) | 0.26 (±0.11) | 0.25 (±0.09) |

U-Net-observer variability resulted in significantly different values from the values of inter-observer variability.

## 4. Discussion

Today, the diagnosis of AD is mainly obtained via clinical evaluation and neuropsychological tests, which show several limitations in terms of the early diagnosis and the objectivity of the final diagnosis.

Several reports suggest that 18F-FDG PET may highlight hypometabolic patterns in brain glucose metabolism typical of Alzheimer's disease.

Machine learning (ML), and DL algorithms, in particular, have found increasing utility in Alzheimer's studies from PET imaging [25].

AD diagnosis methods, based on CNN architecture from 18F-FDG PET imaging data have been proposed [15], resulting in the automatic classification of AD and MCI subjects; however, these methods can not specify which regions are affected by the pathology, i.e., brain areas where hypometabolism occurs. As far as we know, ML models have never been used for the automatic segmentation of diagnostic regions of AD, and this work is the first to use DL techniques to achieve this objective.

Defining hypometabolic areas is more complicated than identifying single regions covering organs or tumor regions, differing from other tissues with grey level, texture, gradient, borders, and shape. In addition, a further challenge is the need to build the learning base for the supervised analysis of DL applications by manually processing large volumes of 3D data.

This work compared different CNNs for automatic 3D volume segmentation; the proposed networks were derived from the U-Net3D and V-Net architectures, which are the best-known models for medical image segmentation. For V-Net, two models with a different number of network levels were implemented: V-Net and V-Net-TwoLevel. For U-Net3D, which has a simpler architecture than V-Net, three models were considered: U-Net3D, U-Net3D-NoMaxPoolingThirdDim, and U-Net3D-TwoLevel.

The cross-validation results (Table 2) showed a better performance in U-Net3D-based networks; in particular, the U-Net3D model showed higher values of both *DSC* and validation *DSC* than the other U-Net3D-based and V-Net networks.

Additionally, in terms of a more qualitative interpretation of the results, as shown in Figure 4, by comparing the ROIs detected by the different networks with respect to the ground truth (second column in Figure 4), we found that the U-Net3D-based architectures were significantly better than V-Net-based ones, both in the segmentation of regions and edge details.

The quantitative average values of the indices (*DSC*, *AOC*, and *EAC*) for each labeled brain region and each architecture are given in Table 3. A comparison of the values obtained with V-Net and U-Net3D showed an improvement of 50% (from about 0.50 to about 0.75) of *AOC*, which was verified. This demonstrates that the choice of U-Net3D-based networks, which has a simpler architecture and need to learn a smaller number of parameters than V-Net, is best for this aim and for this dataset. Among the U-Net3D-based networks, the best performance was obtained with U-Net3D, with a *DSC* index in the range of 0.75–0.77 and *AOC* in the range of 0.71–0.75 for each label. The *EAC* of U-Net3Ds had a slightly higher mean value than the *EAC* of U-Net3D-TwoLevel, demonstrating a little overestimation of segmented regions; however, given the variability of the data (i.e., the standard deviation values), the *EAC* values for the two networks may be considered to be very similar.

The boxplot in Figure 5 shows that, in U-Net3D, there were no significant differences in the index values between the segmented regions; this confirmed the fact that the net was able to clearly define all the regions of interest, even where there was low tracer emission, i.e., with the presence of the pathology.

The best-proposed architecture (U-Net3D) was able to automatically identify all the regions of interest, both on the control subjects, as shown in Figure 6a, and on MCI patients, as shown in Figure 6b, as well as on AD patients, as shown in Figure 6c. This means that the proposed model is not influenced by the presence of a hypometabolic region and determines automatic segmentation comparable for each subject class (i.e., control, AD, or MCI).

The results regarding the inter-observer variability (shown in Table 4) demonstrated that the overlap value between the regions drawn by observer one and those determined by observer two (*DSC* index and *AOC* index) was lower than the *DSC* and *AOC* values between the proposed automatic method and the ground truth (*DSC*, on average, equal to 0.62 and *AOC*, on average, equal to 0.63 in the inter-observer variability analysis; *DSC* equal to 0.76 and *AOC* equal to 0.73 for the automatic ground-truth comparison). As regards the *EAC* index, the inter-observer variability was very similar to that obtained from the comparative analysis between the proposed automatic method and the ground truth.

These results demonstrated a statistically significant difference between the inter-observer variability and the automatic ground-truth comparison. This leads us to state that the ability of the proposed method to perform automatic segmentation is comparable to or superior to that of a manual segmentation method, with the advantage that automatic segmentation is faster, less tedious, and less prone to errors.

The proposed method is the first that obtains the automatic segmentation of cerebral areas from 18F-FDG PET 3D volumes. Previous works are related to automatic brain segmentation of volumetric data but are based on MRI techniques. The MRI segmentation approaches, as described in [26], use typical DL architectures commonly used for image segmentation. Despite the significant impact of these DL techniques, it is still challenging to have a generic method to apply to these imaging techniques, and their performance highly depends on several steps such as preprocessing, initialization, and postprocessing of the dataset, thus making training complicated.

The method proposed in this work, compared with the methods applied to MRI data, has the following two main advantages for PET images: (i) the U-Net-based model is much faster to train due to its context-based learning; (ii) the optimization of hyperparameters and the choice of the optimum number of layers of the U-Net architecture allows one to obtain *DSC* values comparable to the ones obtained from MRI data but with a simpler architecture.

The U-Net architectures in the literature are mainly used to detect tumors or nodules that are identifiable from other tissue, and their localization is usually concentrated in a single area. In this paper, we proved that U-Net3D-based architectures are able to recognize different cerebral regions distributed throughout the 3D volume, and they can be obtained both for normal and hypometabolic regions.

The present work has some limitations. The first one is the limited volume of data. However, it is known that obtaining very high volumes of case study data from biomedical images is very difficult, and for PET images, it is even more complicated, due to the type of the methodology, which is ionizing; therefore, the method is used only after careful cost–benefit evaluation. This problem could be overcome by carrying out studies involving a network of nuclear medicine centers, thus increasing the volume of data. The use of the proposed U-Net3D CNN on the data from other centers is the goal of a future study.

Another reason why the availability of a large number of datasets was reduced is that to have the ground truth, it is necessary to manually select all the brain regions of interest for diagnosis, even several times to have an inter-operator comparison, and this takes a considerable amount of manpower time.

The greatest difficulty for the network to correctly select the ROIs is due to the fact that the number of slices covering a particular region of interest differed between the datasets, which can make learning more complicated for the network. Moreover, the more frequent errors in defining the ROIs occurred when passing from the slices that included four labels (frontal lobes and temporal lobes) than when passing from the ones with two labels (temporal lobes), also because the temporal lobes were generally identified in only 2–3 slices of the entire volume, so the network had fewer examples from which to learn.

However, it should be noted that the main objective of the present work was to perform a preliminary feasibility study, which would allow an evaluation of whether it was possible to perform automatic segmentation on 18F-FDG brain images from subjects in different stages of AD.

## 5. Conclusions

The proposed method for the automatic segmentation of 3D brain PET images, based on a convolutional neural network, proved to be very promising to allow an accurate diagnosis of the presence of Alzheimer's disease, even in the most critical situations, such as MCI. In fact, once the segmentation was performed using the U-Net3D-based net, the emission from the segmented region, and therefore the extent of metabolic activation, was automatically evaluated, constituting a great aid for the physician in the localization of hypometabolic regions for an accurate diagnosis of AD.

**Author Contributions:** Conceptualization, A.G. and D.G.; methodology, E.P., M.F.S. and V.P.; software, E.P. and L.A.D.S.; formal analysis, E.P. and D.G.; resources, A.G., D.G. and C.R.; data curation, C.R.; writing—original draft preparation, E.P.; writing—review and editing, M.F.S. and V.P.; supervision, M.F.S. All authors have read and agreed to the published version of the manuscript.

**Funding:** This research received no external funding.

**Institutional Review Board Statement:** The study was conducted according to the guidelines of the Declaration of Helsinki and approved by the Institutional Ethics Committee.

**Informed Consent Statement:** Informed consent was obtained from all patients involved in the study.

**Data Availability Statement:** Data are available on request due to privacy restrictions.

**Conflicts of Interest:** The authors declare that they have no conflict of interest.

## Abbreviations

| | |
|---|---|
| **AD** | Alzheimer's Disease |
| **PET** | Positron emission tomography |
| **MRI** | Magnetic resonance imaging |
| **18F-FDG PET** | 18F-fluorodeoxyglucose positron emission tomography |
| **SUV** | Standardized uptake value |
| **ML** | Machine learning |
| **CNNs** | Convolutional neural networks |
| **U-Net3D** | Convolutional neural network for 3D image segmentation with U-shaped architecture |
| **V-Net** | Convolutional neural network for 3D image segmentation with V-shaped architecture |
| **CT** | Computed tomography |
| **RM** | Magnetic resonance |
| **DL** | Deep learning |
| **MCI** | Mild cognitive impairment |
| **ROI** | Regions of interest |
| *DSC* | Dice similarity coefficient |
| *AOC* | Overlapping area coefficient |
| *EAC* | Extra area coefficient |

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
