# Peer review of "Convolution Neural Networks for the Automatic Segmentation of 18F-FDG PET Brain as an Aid to Alzheimer’s Disease Diagnosis"

_electronics, doi:10.3390/electronics11142260_

Round 1
Reviewer 1 Report
The work presents convolution neural networks (CNNs) for an automatic segmentation of 18F[FDG]PET brain scans to aid the diagnosis of Alzheimer’s Disease (AD). The paper uses existing UNet architectures to test them on their data, so from algorithmic side there is not much novelty. Having that said, the authors should include more related work and intensively discuss/compare their work with the related work.
The manuscript needs to be proof read by a native speaker, e.g. in the Abstract already:
… on which was performed manual segmentation by a expert to obtain ground truth …
-> … on which a manual segmentation was performed by aN expert to obtain the ground truth …
The images in Figure 1. look deformed to me (stretched in y-direction).
Author Response
Please, see attached file

Reviewer 2 Report
Hi
This paper deals with the use of neural networks in the field of medicine, a good topic. But after studying the article, I found some shortcomings, including:
1. The introduction must contain studies completed in this field, as it must contain 70 percent of the total references of the paper.
2. What is the main contribution of the paper? And other goals that have been achieved and achieved in this work?
3. In the summary, what do the following abbreviations mean: MCI, U-Net3D, EAC, AOC, DSC, V-net.
4. Figures 2 and 3 are not clear.
5. Where exactly are neural networks used? Why was it used? And how was it used?
6. What are the pros and cons of using neural networks in the paper?
7.Is Figure 6 taken empirically? Is it a test sample? How was it obtained? I want a detailed explanation of how it was obtained.
8. Is it possible to complete a comparative study with other works on the same subject?.
9. What are the characteristics of neural networks used? How many inner, outer and hidden layers are there?
Author Response
Please, attached file

Round 2
Reviewer 1 Report
no further comments
Reviewer 2 Report
Thanks for the edits made.